# Arsenic Induces M2 Macrophage Polarization and Shifts M1/M2 Cytokine Production via Mitophagy

**DOI:** 10.3390/ijms232213879

**Published:** 2022-11-10

**Authors:** Chih-Hsing Hung, Hua-Yu Hsu, Hsin-Ying Clair Chiou, Mei-Lan Tsai, Huey-Ling You, Yu-Chih Lin, Wei-Ting Liao, Yi-Ching Lin

**Affiliations:** 1Research Center for Environmental Medicine, Kaohsiung Medical University, Kaohsiung 807, Taiwan; 2Department of Pediatrics, Kaohsiung Municipal Siaogang Hospital, Kaohsiung 812, Taiwan; 3Department of Pediatrics, Kaohsiung Medical University Hospital, Kaohsiung Medical University, Kaohsiung 807, Taiwan; 4Graduate Institute of Medicine, College of Medicine, Kaohsiung Medical University, Kaohsiung 807, Taiwan; 5Department of Pediatrics, School of Medicine, College of Medicine, Kaohsiung Medical University, Kaohsiung 807, Taiwan; 6Department of Biotechnology, College of Life Science, Kaohsiung Medical University, Kaohsiung 807, Taiwan; 7Teaching and Research Center, Kaohsiung Municipal Siaogang Hospital, Kaohsiung 812, Taiwan; 8Department of Laboratory Medicine, Kaohsiung Chang Gung Memorial Hospital, Kaohsiung 833, Taiwan; 9Division of General Internal Medicine, Department of Internal Medicine, Kaohsiung Medical University Hospital, Kaohsiung Medical University, Kaohsiung 807, Taiwan; 10Department of Medical Humanities and Education, School of Medicine, College of Medicine, Kaohsiung Medical University, Kaohsiung 807, Taiwan; 11Department of Medical Research, Kaohsiung Medical University Hospital, Kaohsiung, Medical University, Kaohsiung 807, Taiwan; 12Department of Laboratory Medicine, Kaohsiung Medical University Hospital, Kaohsiung Medical University, Kaohsiung 807, Taiwan; 13Doctoral Degree Program in Toxicology, College of Pharmacy, Kaohsiung Medical University, Kaohsiung 807, Taiwan; 14Department of Laboratory Medicine, School of Medicine, College of Medicine, Kaohsiung Medical University, Kaohsiung 807, Taiwan

**Keywords:** arsenic, macrophage polarization, mitophagy, epithelial–mesenchymal transition, quinine, coculture, macrophage

## Abstract

Arsenic is an environmental factor associated with epithelial–mesenchymal transition (EMT). Since macrophages play a crucial role in regulating EMT, we studied the effects of arsenic on macrophage polarization. We first determined the arsenic concentrations to be used by cell viability assays in conjunction with previous studies. In our results, arsenic treatment increased the alternatively activated (M2) macrophage markers, including arginase 1 (*ARG-1*) gene expression, chemokine (C-C motif) ligand 16 (CCL16), transforming growth factor-β1 (TGF-β1), and the cluster of differentiation 206 (CD206) surface marker. Arsenic-treated macrophages promoted A549 lung epithelial cell invasion and migration in a cell co-culture model and a 3D gel cell co-culture model, confirming that arsenic treatment promoted EMT in lung epithelial cells. We confirmed that arsenic induced autophagy/mitophagy by microtubule-associated protein 1 light-chain 3-II (LC3 II) and phosphor-Parkin (p-Parkin) protein markers. The autophagy inhibitor chloroquine (CQ) recovered the expression of the inducible nitric oxide synthase (*iNOS*) gene in arsenic-treated M1 macrophages, which represents a confirmation that arsenic indeed induced the repolarization of classically activated (M1) macrophage to M2 macrophages through the autophagy/mitophagy pathway. Next, we verified that arsenic increased M2 cell markers in mouse blood and lungs. This study suggests that mitophagy is involved in the arsenic-induced M1 macrophage switch to an M2-like phenotype.

## 1. Introduction

Arsenic is an environmental pollutant present in groundwater, soil, and air in the environment that threatens the health of millions in the world population. Long-term exposure to environmental arsenic brings damage to organs such as the lung [1]. Previous studies have shown that arsenic exposure is associated with lung cancer [2], disrupted lung respiratory function [3], non-malignant lung diseases [4], asthma [5], and pulmonary fibrosis [6,7].

Macrophages are major airway immunocytes in lung immune homeostasis that function by phagocytosis and chemokine secretion in responding to antigens [8]. Two distinct states of macrophages have been recognized: the classically activated (M1) macrophage phenotype and the alternatively activated (M2) macrophage phenotype [9]. While macrophages are typically expected to polarize toward the M1 phenotype in response to T-helper 1 (Th1) cytokine interferon-γ, M2 polarization was initially discovered as a response to the T-helper 2 (Th2) cytokine interleukin (IL)-4 [10]. M1 macrophages express inducible nitric oxide synthase (iNOS) and Th1 cell-attracting chemokines such as chemokine (C-X-C motif) ligand 9 (CXCL9) and CXCL10 [11]. M2 macrophages are characterized by the expression of arginase-1 receptor (ARG-1), chemokine (C-C motif) ligand 1 (CCL1), CCL16, and CCL22 [12,13]. Cells of the monocyte/macrophage lineage are known to be diverse, with high plasticity.

Epithelial–mesenchymal transition (EMT) is characterized by the loss of the epithelial phenotype and the acquisition of the mesenchymal phenotype. EMT is important in regulating the invasion and migration of tumor cells and the development and wound repair processes. EMT is a universal process in lung diseases that promotes tissue remodeling and fibrosis [14,15]. Several studies have revealed that EMT is the critical mechanism by which arsenic induces pulmonary diseases [16,17,18]. Extracellular factors/stimuli may induce EMT [19]. A previous study has reported that macrophages secrete transforming growth factor-β1 (TGF-β1), a factor involved in matrix remodeling and tissue remodeling, and trigger EMT [20].

Exposure to highly reactive oxygen species (ROS) causes mitochondrial damage, and acute exposure to a high dose of arsenic would result in ROS overproduction [21]. Mitophagy, which is mitochondrial autophagy, usually happens in defective mitochondria following injury or stress, and microtubule-associated protein 1 light-chain 3 (LC3) is one of the components of the autophagy pathway [22,23]. The signaling pathway mediated by PTEN-induced putative kinase 1 (PINK1) and the E3 ubiquitin ligase Parkin has been reported to induce mitophagy in mammalian cells [24]. ROS-induced mitophagy in alveolar macrophages would contribute to pulmonary fibrosis [25]. Arsenic-induced mitophagy has been reported in hepatic and pancreatic cells. The toxic mechanisms activated by arsenic in alveolar macrophages are not fully understood. In this study, we aimed to evaluate the effects of mitophagy in macrophages and provide potential therapeutic strategies for treating arsenic-induced lung damage.

## 2. Results

### 2.1. Arsenic Induced M2 Polarization and Increased M2c-Asscoated Cytokines/Chemokines in m1 Macrophages

For selecting the arsenic treatment concentration, we performed cytotoxicity tests using XTT assays. As illustrated in Figure 1A, an arsenic concentration less than 10 μM exhibited no significant inhibition of cell viability in phorbol 12-myristate 13-acetate (PMA)-primed macrophage (M0), THP-1-drived M1, and THP-1-drived M2 macrophages after 24 h of incubation. In addition, in chronic exposure, the arsenic concentration range was between 0.1 and 0.5 mg/kg, approximately equal to a range of 1.4 to 6.9 μM [26]; therefore, arsenic concentrations of 1 and 5 μM were applied during the experiments.

Cluster of differentiation 80 (CD80) was used as an M1 macrophage marker, and cluster of differentiation 206 (CD206) was used as an M2 macrophage marker in this study. Confocal laser microscopy showed that the treatment with arsenic (1 and 5 μM) decreased CD80 and increased CD206 accumulation in both PMA-primed macrophages (M0) and THP-1-drived M1 macrophages (Figure 1B,C). CD206 expression was decreased in THP-1-drived M2 macrophages after treatment with 5 μM arsenic for 24 h (Figure 1D).

Then, we examined the effects of arsenic on M1/M2 chemokine production. Arsenic significantly decreased the production of the M1-related chemokine CXCL-10 (Figure 2A) and the cytokine interleukin (IL)-1β (Figure 2B) in THP-1-drived M1 macrophages in a concentration-dependent manner. Arsenic significantly decreased the production of the M2a-related chemokine CXCL-22 (Figure 2C) and the M2b-related chemokine CCL-1 (Figure 2D) in THP-1-drived M2 macrophages in a concentration-dependent manner. Arsenic significantly increased the production of the M2c-related cytokine IL-10 (Figure 2E) in THP-1-derived M1 macrophages and increased the M2c-related chemokine CCL-16 (Figure 2F) in both THP-1-drived M1 and M2 macrophages in a concentration-dependent manner. These data indicated that arsenic was associated with a shift in macrophage polarization, from M1 to M2. Arsenic promoted M2 macrophage polarization and increased M2c-related cytokine/chemokine production.

### 2.2. An Autophagy Inhibitor Reduced Arsenic-Induced M2 Polarization

Inducible nitric oxide synthase (iNOS) is an important marker for M1 macrophage differentiation, and Arginase 1 (ARG-1) is a marker for M2 macrophages. Pretreatment with arsenic (1 and 5 μM) for 24 h reduced the expression of the *iNOS* gene in THP-1-drived M1 macrophages (Figure 3A). Pretreatment with arsenic (1 and 5 μM) for 24 h increased *ARG-1* gene expression in THP-1-drived M1 macrophages but decreased *ARG-1* gene expression in THP-1-drived M2 macrophages (Figure 3B). TGF-β1 secreted by M2 macrophages enhanced cell stemness and migration. Pretreatment with arsenic (1 and 5 μM) for 24 h increased TGF-β1 gene expression in both THP-1-drived M1 and M2 macrophages (Figure 3C). However, the arsenic-induced effects on *iNOS, ARG-1* and TGF-β1 were reversed by the autophagy inhibitor chloroquine (IL) and the antioxidant N-acetylcysteine (NAC) in THP-1-drived M1 macrophages (Figure 3D–F). These data indicated that ROS-related autophagy might be involved in arsenic-induced M2 macrophage polarization.

### 2.3. Arsenic Promoted Autophagy in M1 Macrophages

Mitophagy often occurs in defective mitochondria following damage or stress. Since the autophagy inhibitor CQ and the antioxidant NAC reversed arsenic-promoted M2 macrophage polarization, we further examined whether arsenic could increase autophagy-related protein LC3-II expression in THP-1-drived M1 macrophages. Arsenic also induced p62/sequestosome 1 (SQSTM1) protein degradation and promoted full-length PINK1 (63 kd) stabilization in M1 macrophages. In addition, the mitophagy-related protein phosphor-Parkin was increased by arsenic treatment in in THP-1-drived M1 macrophages (Figure 4A). Confocal laser microscopy also showed that arsenic treatment increased LC3 (Figure 4C, green) accumulation in mitochondria (Figure 4C, red) in THP-1-drived M1 macrophages, which is a typical phenomenon associated with mitophagy. The PINK1–Parkin axis is a prominent mitophagy regulatory pathway. When we knocked down PINK1 gene expression by ShRNA, we found that arsenic-induced Arg-1 expression and TGF-β1 release in M1 macrophage were reversed (Figure 4E–G). These findings suggested mitophagy was involved in arsenic-induced M2-like changes.

### 2.4. Arsenic-Treated M1 Macrophages Promoted Epithelial Cell Migration and Invasion

Cell migration is a critical property of EMT. The effects of macrophages on arsenic-induced cell migration of lung epithelial cells were determined by Transwell assays in this study. As shown in Figure 5A,B, A549 cells co-cultured with THP-1-drived macrophages showed significantly enhanced cell migration compared with A549 cells alone under a 5 μM arsenic treatment.

We further investigated whether arsenic could induce macrophages to enhance the invasive ability of lung epithelial cells through a 3D gel. THP-1-drived M1 macrophages were cultured in the lower layer of the gel, and A549 cells were cultured in the upper layer. After arsenic treatment, the gel underwent sectioning and H&E staining. A shown in Figure 5C, more A549 cells invaded the 3D gel under arsenic treatment (1 and 5 μM) in a concentration-dependent manner.

### 2.5. Arsenic-Treated Mice Showed Increased M2 Macrophages

To confirm the arsenic-induced M2 changes in vivo, we treated C57BL/6 mice (8 weeks old) with 50 mg/L of arsenic-containing drinking water for 24 weeks. After treatment, whole blood was collected for CD80 (M1 marker) and CD206 (M2 marker) flow cytometry analysis, and the lung tissue was collected for CD206 staining. The results showed that the number of CD206+ cells was increased in arsenic-treated mouse blood (Appendix A). In arsenic-treated mice, the airway epithelial thickness was increased, and the number of infiltrated CD206 + cells in lung tissue was increased (Appendix A).

## 3. Discussion

Arsenic has been classified as a Group 1 human carcinogen by the International Agency for Research on Cancer (IARC), and IARC has established a causal role for arsenic in lung cancer. It is known that arsenic has a bi-phase effect on cell viability. In general, a concentration of arsenic lower than 10 µM showed growth-promoting effects and a concentration higher than 10 µM showed cytotoxic effects [27,28]. In the current study, we observed that 5 µM arsenic increased the viability of M0, M1 and M2 cells, suggesting that non-toxic concentrations of arsenic activated the macrophages. However, this enhanced cell viability was decreased under 10 µM arsenic treatment in M0 cells, but not in M1 and M2 cells. M0 cells are PMA-treated THP-1 monocytes, which are pre-mature macrophages. It is possible that pre-mature M0 macrophages have a reduced response to arsenic, compared to M1 and M2 macrophages.

The lung is a usual site of arsenic exposure, which can result in toxicity. Arsenic is not only a lung carcinogen; in fact, its non-malignant lung effects were also discussed recently [29]. Arsenic exposure will cause inflammation, while long-term exposure to arsenic can cause chronic inflammation [30,31]. Ingested arsenic is associated with non-malignant pulmonary diseases such as bronchitis, asthma, and pneumonia [32,33,34]. Increased levels of serum Immunoglobin E and reduced lung function parameters have been reported in populations exposed to arsenic [5,35]. Several studies have also suggested that arsenic exposure in utero and in childhood is associated with an increased risk of respiratory effects [36,37,38].

Arsenic exposure may cause pulmonary and systemic inflammation and an altered immune response. Prolonged ingestion of arsenic induced elevated levels of pro-inflammatory mediators, such as tumor necrosis factor-α (TNF-α), IL-8, IL-6, and IL-12 in the blood, and excessive ROS generation in the airways [30]. Arsenic promoted the polarization of M2 macrophages [39]. In this study, high-dose arsenic (1 and 5 μM) decreased *iNOS* and increased *ARG-1* gene expression in a dose-dependent manner. These findings support the conclusion that arsenic exposure shifted macrophage polarization from an M1-like to an M2-like phenotype. Moreover, arsenic decreased the production of M1-related chemokines (CXCL-10 and IL-1β) in THP-1-derived M1 macrophages and decreased the production of the M2a chemokine (CCL22) and the M2b chemokine (CCL1) in THP-1-derived M2 macrophages. Otherwise, arsenic increased the production of the M2c chemokine (CCL16) in both THP-1-derived M1 and M2 macrophages. These findings support the conclusion that arsenic exposure facilitated M2 macrophage polarization and increased M2c-related chemokine CCL-16 production. TGFβ1 regulates cell proliferation, growth, differentiation, and cell migration. TGFβ1 also has immunomodulatory and profibrogenic effects [40]. A common finding in fibrotic processes in the lung is an increase in TGF-β levels, which is proportionate in many cases to the degree of fibrosis that is generated [41]. Arsenic exposure increased TGF-β1 in THP-1-derived M1 and M2 macrophages in this study.

It has been reported that *ARG-1* is highly expressed in M2a macrophages, and TGF-β1 is highly expressed in both M2a and M2c macrophages [42]. Our results showed that arsenic decreased *ARG-1* gene expression but increased *TGF-β1* gene expression in M2 macrophages, indicating that arsenic promoted M2c polarization in M2 macrophages. These results suggest that arsenic promoted M2c polarization in M2 macrophages.

EMT is an essential mechanical change contributing to carcinogenesis, airway remodeling, and tissue fibrosis. Previous studies have discussed the relationship between arsenic exposure and EMT [18,43]. This study used an in vitro co-culture model to investigate the crosstalk between macrophages and epithelial cells under arsenic exposure. Macrophages have a high tolerance to arsenic toxicity. Higher arsenic concentration promoted M2 polarization and increased epithelial cell migration in co-cultures with macrophages.

Increased ROS production is considered a common pathway in arsenic toxicity, and ROS overproduction has been demonstrated to promote EMT progression [44]. The antioxidant NAC and autophagy inhibitor chloroquine reversed these effects, implying oxidative stress-associated autophagy is involved in arsenic-induced M2 macrophage polarization. Autophagy can be considered a critical cellular self-defense mechanism that can protect cells from various environmental stresses, such as inflammation, oxidative stress, and injury. It has been considered that arsenic causes lung damage by inducing a redox imbalance, apoptosis, and an inflammatory response that may cause EMT, resulting in airway dysfunction. The effect of mitophagy on macrophage polarization has been discussed and studied recently [25,45,46,47,48,49]. In the present study, we found arsenic increased the expression of the autophagy protein LC3 II and induced the degradation of the p62/SQSTM1 (p62) protein. The p62 protein is a selective cargo receptor for misfolded proteins in autophagy [50]. It is reported that p62 binds with Parkin and assists the PINK1/Parkin mitophagy pathway [51]. During the proteolysis stage of autophagy or mitophagy, the p62 protein itself is also degraded. Therefore, a decreased level of p62 is a commonly used maker of activated proteolysis [52]. In addition, the level of the mitophagy-associated proteins phospho-Parkin was increased by arsenic treatment in THP-1-derived M1 macrophages. These findings suggest that mitophagy might be involved in arsenic-induced M2 macrophage polarization.

Chloroquine (CQ) is a widely used antimalarial drug. Recently, a study pointed out that CQ has the function of transforming macrophages from M2-like to M1-like macrophages [53]. Inhibition of mitochondrial autophagy will lead to M1 macrophages. This study found that CQ can alter arsenic-induced M2 macrophage polarization. The polarization plasticity of macrophages has been found in many diseases. Subsequent research on the plasticity of polarization regulation in macrophages will considerably shed light on the causes or resolution of some diseases. More research in the future might clarify the principle and mechanisms of action of CQ and other drugs affecting the polarization of macrophages to provide directions for future therapeutic strategies.

In conclusion, arsenic-promoted M2 macrophage polarization might occur via mitophagy. Macrophage polarization changes cause EMT in lung epithelial cells, which might lead to abnormal lung airway tissue repair. Our findings suggest a possible therapeutic strategy to counteract the effects of arsenic on pulmonary airways.

## 4. Material and Methods

### 4.1. Cell Culture and Macrophage Polarization

The human acute monocytic leukemia cell line THP-1 is a model cell line for in vitro monocyte–macrophage differentiation [54]. Our THP-1 cells were cultured in Roswell Park memorial institute (RPMI 1640) medium containing 10% fetal bovine serum (FBS). According to the ATCC protocol, we maintained the cell density at 5 × 10^5^ cells/mL at 37 °C in a CO_2_ incubator. To generate pre-mature macrophages (THP-1-derived M0 macrophages), cultured THP-1 monocytes were treated with PMA (20 ng/mL) for 24 h. To generate THP-1-derived M1 macrophages, cultured THP-1 cells were treated with 20 ng/mL of PMA, 20 ng/mL of IFN-γ, and 20 ng/mL of LPS for 72 h. To generate THP-1-derived M2 macrophages, cultured THP-1 cells were treated with 20 ng/mL of PMA and 20 ng/mL of IL-4 for 72 h [55].

A549 cells, a human adenocarcinoma epithelial cell line, were cultured (5 × 10^5^ cells/mL) in DMEM containing 10% FBS at 37 °C in a CO_2_ incubator.

### 4.2. XTT Cell Viability Assay

M0, M1, or M2 cells (5 × 10^5^ cells/mL) were seeded in 96-well plates and treated with 1–100 μM of arsenic for 24 h. After treatment, 50 μL of the XTT reagent mixture (Roche, Germany) was added into the 96-well plate. After incubating for 8 h at 37 °C, the colored product of XTT was detected by an ELISA reader at 450 nm.

### 4.3. Quantitative Polymerase Chain Reaction (qPCR)

After the treatments, cellular mRNA was isolated using the TRIzol™ reagent (Invitrogen, Carlsbad, CA, USA) according to the manufacturer’s instruction. The isolated mRNA was reverse-transcribed to cDNA using the TaKaRa PrimeScript RT Reagent Kit (Takara Bio Inc., Shimogyō-ku, Kyoto, Japan) according to the manufacturer’s instruction. We used the StepOnePlus™ Real-Time PCR System (Applied Biosystems, Foster City, CA, USA) to amplify the cDNA with the SYBR Green Real-Time PCR Master Mix (Thermo Fisher, Waltham, MA, USA) labeling. PCR protocol: denaturation at 95 °C for 3 s; annealing at 50–67 °C (according to primer Tm) for 30 s; extension at 60 °C for 30 s. After 40 cycles, the samples were heated to 95 °C and cooled down to generate the melting curve to exclude primer dimer effects. The primer sets are reported as Table 1:

### 4.4. Enzyme-Linked Immunosorbent Assay (ELISA)

Human CXCL10, IL-1β, CCL22, CCL1, CCL16, IL-10, and TGF-β1 ELISA kits (Quantikine, R&D System, Minneapolis, MN) were used in this study. Cell-free supernatants from arsenic-treated M0/M1/M2 cells were collected for ELISA according to according to the manufacturer’s instruction. The CXCL10, IL-1β, CCL22, CCL1, CCL16, IL-10, and TGF-β1concentrations were calculated by standard curves with r2 higher than 0.99.

### 4.5. Migration Assay

A549 cells (5 × 10^5^ cells/mL) were seeded in the inserts (pore size 8 μm) of trans-wells, and THP-1-dervided M1 cells (5 × 10^5^ cells/mL) were seeded in the outer well of the trans-wells. These co-cultured cells were treated with 1 or 5 μM arsenic for 24 h. After treatment, the inserts were washed 3 times with PBS and fixed with 3.7% formalin at 37 °C for 5 min. The cells that had migrated through the insert membrane were stained with Giemsa for 15 min at room temperature. After staining, the inserts were washed with PBS for 5–10 times until no excess stain, and pictures were taken under an optical microscope (40×).

### 4.6. Invasion Assay

A549 cells (5 × 10^5^ cells/mL) were seeded on a 2 mg/mL collagen-I gel (Sigma, St. Louis, MO, USA) (2 mg/mL) in the insert (pore size 0.4 μm) of a trans-well. THP-1-dervided M1 cells (5 × 10^5^ cells/mL) were co-cultured in the outer well of the trans-well with or without 1 or 5 μM arsenic. These cells were cultured for 5 days. After treatment, the collagen gels (with A549 cells on it) were formalin- (3.7%) fixed by and paraffin-embedded and then cut in 5 μm sections. The sections were stained by hematoxylin and eosin (H&E) and observed under an optical microscope.

### 4.7. Western Blotting

Western blotting was modified from our previous study [56]. Arsenic-treated M0/M1/M2 cells were incubated in RIPA lysis buffer 10× (20–188, Merck, Germany) with a cocktail of protease inhibitors for 2 h at 4 °C. The protein centration in the cell lysates was measured using the BCA protein quantification kit (Bio-Rad, Hercules, CA, USA). In total, 40 μg of proteins each sample was subjected to 10% sodium dodecyl sulfate polyacrylamide gel electrophoresis (SDS-PAGE). After SDS-PAGE, the proteins in the polyacrylamide gels were transferred to nitrocellulose membranes (Bio-Rad, Hercules, CA, USA). The nitrocellulose membranes were incubated with anti-GAPDH (GTX100118, GeneTex, Irvine, CA), anti-LC3 (Ab192890, Abcam, UK), anti-p62/SQSTM1 (ab91526, Abcam), anti-PINK1 (6946, Cell Signaling, Danvers, MA, USA), or anti-p-Parkin (36866, Cell Signaling) antibodies at room temperature for 12 h and then incubated with anti-Rabbit HRP-conjugated secondary antibodies (7074, Cell Signaling, Danvers, Massachusetts) or anti-Mouse HRP-conjugated secondary antibodies (7076, Cell Signaling, Danvers, Massachusetts) for 2 h at room temperature. After the antibody reactions, the western blotting bands were visualized by a chemiluminescence substrate kit (Pierce, Rockford, IL, USA).

### 4.8. Immunofluorescent Staining

For mitochondria–LC3 co-localization staining, M0/M1/M2 cells (5 × 10^5^ cells/mL) were seeded on glass cover slides and treated with arsenic for 24 h. After PBS washing, the slides were stained with MitoTracker Deep Red reagents (M22426, Thermo Fisher, Waltham, MA, USA) according to the manufacturer’s instruction, for mitochondria staining. Then, the cell slides were fixed in 3.7% paraformaldehyde, permeabilized with 0.2% TritonX-100, and washed 3 times with PBS-Tween wash buffer. After washing, the slides were incubated in 500× diluted pAb anti-LC3B (NB100-2220, Novus bio, Centennial, Colorado, CO, USA). Then, they were washed and incubated with the AlexaFlour488 antibody (A-11001, Thermo Fisher, Waltham, MA, USA) for immunofluorescent staining. The stained cells were observed under a confocal laser microscope.

For CD80/CD206 (M1/M2) staining, M0/M1/M2 cells (5 × 10^5^ cells/mL) were seeded on glass cover slides and treated with arsenic for 24 h. After fixing and washing, the slides were incubated with anti-CD80 (WH0000941M1, SIGMA, Saint Louis, MO, USA) and AlexaFlour488 antibodies (A-11001, Thermo Fisher, Waltham, MA) for 2 h; after washing, they were incubated with anti-CD206 (HPA004114, SIGMA, Saint Louis, MO, USA) and AlexaFlour594 (A-32740), Thermo Fisher, Waltham, MA, USA) antibodies for 2 h. After DAPI counter staining, the stained cells were observed under a confocal laser microscope.

### 4.9. PINK1 Knockdown

According to our previous publication [49], lentiviral particles encoding non-targeting shRNA (#TRCN0000208001), effective shRNA targeting PINK1 3′-UTR (#TRCN0000007097; decreased PINK1 protein expression to about 40%), and effective shRNA targeting coding sequence (#TRCN0000199446; decreased PINK1 protein expression to about 40%) were used in this study. These 3 lentiviral particles did not show off-target effects at 1 multiplicity of infection (1 MOI) in THP-1-derived macrophages^49^. In brief, THP-1-derived M1 macrophages were transduced with 1 MOI of shRNA lentiviral particles. After 3 days of puromycin selection (0.5 μg/mL), the THP-1-derived M1 macrophages and cell-free supernatants were harvested for ARG1/iNOS real-time PCR test and TGF-β ELISA tests, respectively.

### 4.10. Animals and Arsenic Exposure

The Animal Use Protocol (IACUC-109251) was approved by the Kaohsiung Medical University-Institutional Animal Care and Use Committee. C57BL/6 mice aged 6–8 weeks were purchased from the National Laboratory Animal Center (Taiwan). Mice were exposed to 50 mg/L (equals 28.8 ppm) of NaAsO2 (Sigma, St. Louis, MO, USA, purity: 99.0%) in drinking water for 24 weeks. The control group did not receive any treatment. After treatment, the whole blood was collected for flow cytometry analysis.

### 4.11. Statistical Analysis

The values presented are the means ± SD; the nonparametric Kruskal–Wallis statistical test, followed by Dunn’s post-test, was performed. *p*-values of < 0.05 were considered statistically significant.

## Figures and Tables

**Figure 1 ijms-23-13879-f001:**
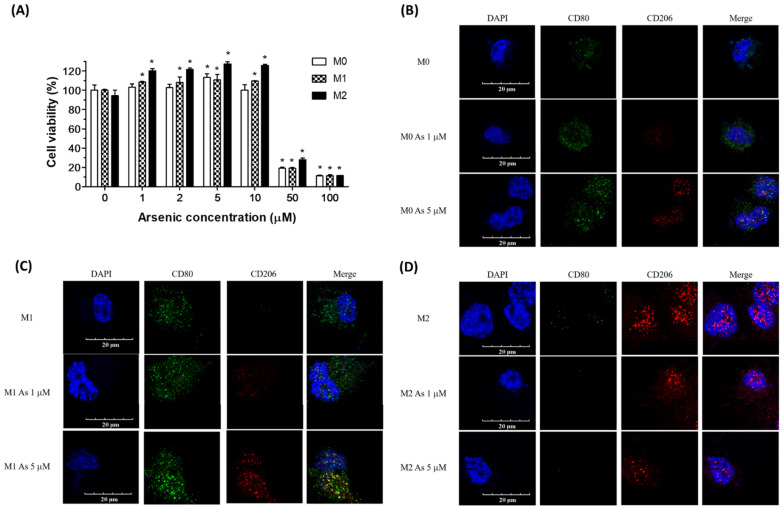
**Non-toxic concentrations of arsenic increase CD206 + M2 macrophages.** (**A**) THP-1 derived M0, M1, M2 macrophages were treated with 1–100 μM of arsenic for 24 h, and cell viability (% of M0 control) was analyzed by the XTT assay. * *p* < 0.05, arsenic-treated vs. untreated M0 cells, arsenic-treated vs. untreated M1 cells, or arsenic-treated vs. untreated M2 cells, by the Kruskal–Wallis test, (N = 3). (**B**–**D**) THP-1-derived M0, M1, M2 macrophages were treated with 1 or 5 μM arsenic for 24 h. The treated cells were stained for CD80 (M1 surface marker, green fluorescence), CD206 (M2 surface marker, red fluorescence), and DAPI (DNA counterstain, blue fluorescence). The cell images were observed under a fluorescence microscope (200×) (N = 3).

**Figure 2 ijms-23-13879-f002:**
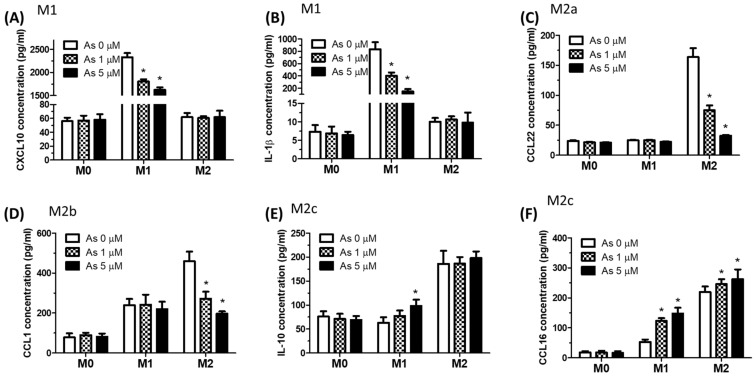
**Arsenic increases M2c-asscoated cytokines/chemokines.** Supernatants from arsenic-treated M0/M1/M2 M1 cells (24 h) were collected for testing: (**A**) M1 chemokine CXCL10, (**B**) M1 cytokine IL-1β, (**C**) M2a chemokine CCL22, (**D**) M2b chemokine CCL1, (**E**) M2c chemokine CCL16, and (**F**) general M2 chemokine IL-10. These chemokines/cytokines were tested by ELISA. * *p* < 0.05, arsenic-treated vs. untreated, by the Kruskal–Wallis test (N = 6).

**Figure 3 ijms-23-13879-f003:**
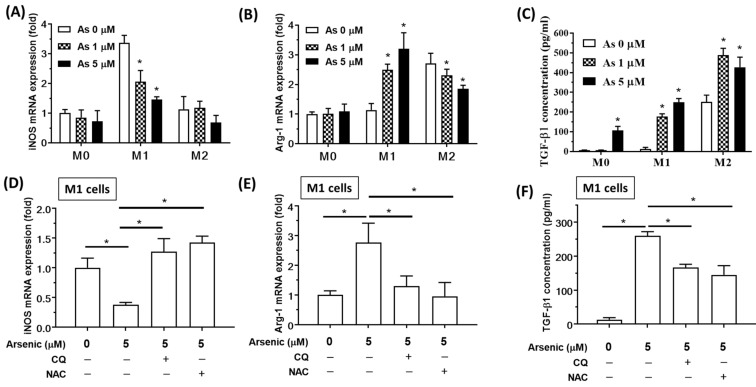
**An autophagy inhibitor reduces arsenic-induced M2 polarization.** THP-1-derived M0, M1, M2 macrophages were treated with 1 or 5 μM arsenic for 24 h. (**A**) The expression of the M1 marker gene iNOS was analyzed by real-time PCR. (**B**) The expression of the M2 marker gene ARG-1 was detected by real-time PCR. (**C**) The release the of M2-associated EMT regulator TGF-β1 as detected by ELISA. * *p* < 0.05, arsenic-treated vs. untreated, by the Kruskal–Wallis test (N = 6). (**D**) THP-1-derived M1 cells were treated with 5 μM arsenic for 24 h, with or without the autophagy inhibitor CQ (10 μM) or the antioxidant NAC (10 μM). After the treatments, iNOS expression was analyzed by real-time PCR. (**E**) THP-1-derived M1 cells were treated with 5 μM arsenic for 24 h, with or without CQ/NAC. After the treatments, ARG-1 expression was analyzed by real-time PCR. (**F**) THP-1-derived M1 cells were treated with 5 μM arsenic for 24 h, with or without CQ/NAC. After the treatments, TGF-β1 concentration in the supernatants was detected by ELISA. * *p* < 0.05 by the Kruskal–Wallis test (N = 5).

**Figure 4 ijms-23-13879-f004:**
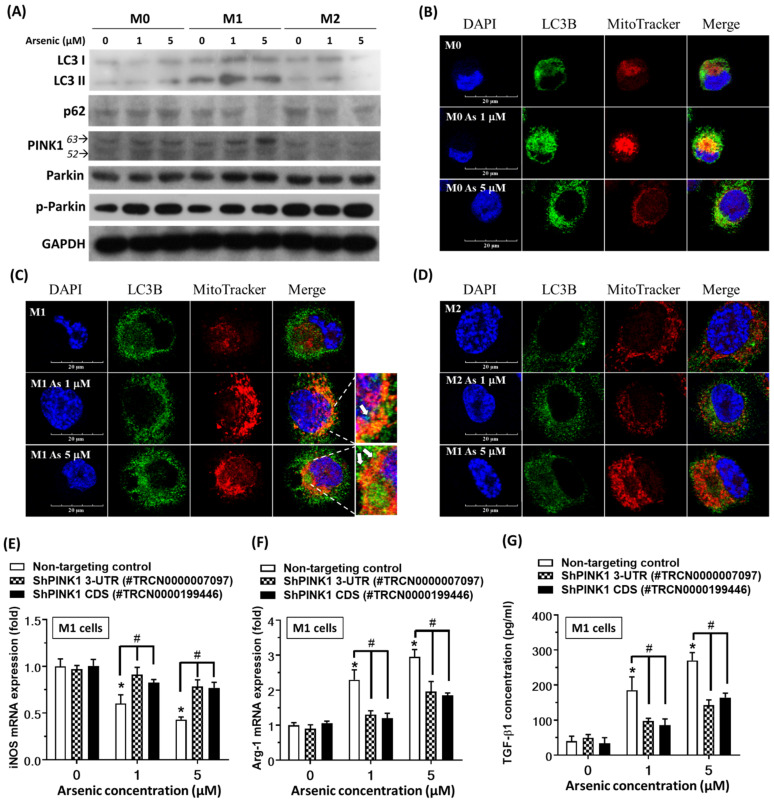
**Arsenic promotes autophagy in M1 macrophages.** THP-1-derived M0, M1, M2 macrophages were treated with 1 or 5 μM arsenic for 24 h. (**A**) The expression of the autophagy-associated proteins LC3 II, p62, PINK1 and phospho-parkin (p-Parkin) was detected by Western blotting (N = 5). (**B**–**D**) THP-1-derived M0, M1, M2 macrophages were treated 1 or 5 arsenic for 24 h and stained with LC3 (green fluorescence), Mitotracker (red fluorescence), and DAPI (DNA counterstain, blue fluorescence). The cell images were observed under a fluorescence microscope (200×) (N = 3). White arrow: LC3 and Mitotracker co-localization (yellow-orange fluorescence). (**E**–**G**) THP-1-derived M1 macrophages were treated with shRNA against PINK1 mRNA 3′-UTR or coding sequences (CDS). After PINK1 knockdown, arsenic-induced iNOS and ARG-1 mRNA expression were detected by real-time PCR. Arsenic-induced TGF-β1 release was detected by ELISA (N = 3). * *p* < 0.05 Non-targeting control with or without arsenic treatments, by the Kruskal–Wallis test; ^#^
*p* < 0.05 Arsenic-treated cells with or without PINK1 ShRNA, by the Kruskal–Wallis test.

**Figure 5 ijms-23-13879-f005:**
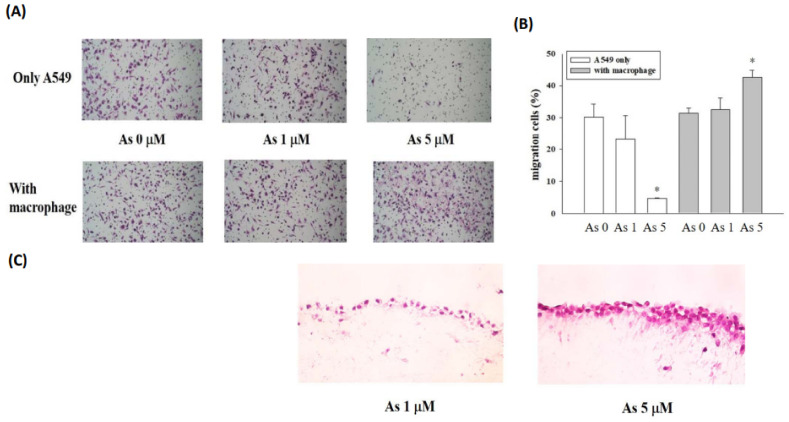
Arsenic-treated M1 macrophages promoted epithelial cell migration and invasion. (**A**) THP-1-derived M1 macrophages were treated with 5 μM arsenic and co-cultured with A549 lung cancer cells in trans-wells for 24 h. The migration of A549 cells through the trans-well membrane was observed upon Giemsa staining (40× under an optical mi-croscope). (**B**) The number of migrated cells was analyzed using ImageJ software. * *p* < 0.05, arse-nic-treated vs. untreated (N = 3). (**C**) A549 cells were cultured on a collagen gel (2 mg/mL) and co-cultured with THP-1-derived M1 macrophages in trans-wells, with or without 1 or 5 μM arse-nic. After 24 h of co-culture, the collagen gels with the A549 cells were fixed, and 5 μM parafilm sections were prepared. After H&E staining, the sections were observed under an optical micro-scope (100×).

**Table 1 ijms-23-13879-t001:** The primer sets of quantitative polymerase chain reaction.

Name	Primer Sequences	Amplicon Size (bp)	Tm
*Human GAPDH-F*	CCACTCCTCCACCTTTGAC	101	53.2
*Human GAPDH-R*	ACCCTGTTGCTGTAGCCA	101	50.3
*Human ARG-1-F*	TCATCTGGGTGGATGCTCACAC	130	67.3
*Human ARG-1-R*	GAGAATCCTGGCACATCGGGAA	130	67.4
*Human iNOS-F*	CGGTGCTGTATTTCCTTACGAGGCGAAGAAGG	259	65.7
*Human iNOS-R*	GGTGCTGCTTGTTAGGAGGTCAAGTAAAGGGC	259	65.7
*Mouse-GAPDH*	GTGTTCCTACCCCCAATGTGT	247	65.2
*Mouse-GAPDH*	ATTGTCATACCAGGAAATGAGCTT	247	63.6
*Mouse-iNOS*	ATGGACCAGTATAAGGCAAGC	428	52.4
*Mouse-iNOS*	GCTCTGGATGAGCCTATATTG	428	52.4
*Mouse-Arg-1*	AGCACTGAGGAAAGCTGGTC	111	65.2
*Mouse-Arg-1*	CAGACCGTGGGTTCTTCACA	111	65

Tm: melting temperature.

## Data Availability

The data presented in this study are available on request from the corresponding author. Please send data requests to: Wei-Ting Liao, Ph.D. Department of Biotechnology, College of Life Science, Kaohsiung Medical University.

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
