# Peer review of "Arsenic Induces M2 Macrophage Polarization and Shifts M1/M2 Cytokine Production via Mitophagy"

_ijms, 2022, doi:10.3390/ijms232213879_

Round 1
Reviewer 1 Report (Previous Reviewer 3)
The authors have dully answered the concerns raised by the reviewer. The current form can be accepted for publication in IJMS.
Author Response
Reviewer 1
Comments and Suggestions for Authors
The authors have dully answered the concerns raised by the reviewer. The current form can be accepted for publication in IJMS.
Reply: Thank you!

Reviewer 2 Report (Previous Reviewer 2)
The manuscript is well written, some minor improvements are needed.
Figure 1. It is not entirely clear why there is an asterisk above MO at 50 µm. Specify what is compared with what.
Line 100. The authors state that Arsenic induced p62/SQSTM1 protein degradation. However, the role of p62 in mitophagy is not discussed anywhere, neither in the introduction nor in the discussion. It is worth adding a discussion with relevant references.
Please add the full name to the some abbreviated terms at the first mention.
Author Response
Reviewer 2
Comments and Suggestions for Authors
The manuscript is well written, some minor improvements are needed.
- Figure 1. It is not entirely clear why there is an asterisk above MO at 50 µm. Specify what is compared with what.
Reply: The asterisk above M0 at 50 μM indicated P<0.05 between arsenic treated vs un-treated in M0 cells. Thank you very much for this question. We have explained the statistical comparison more detailly in Figure legends. Please refer to lines 72-73.
- Line 100. The authors state that Arsenic induced p62/SQSTM1 protein degradation. However, the role of p62 in mitophagy is not discussed anywhere, neither in the introduction nor in the discussion. It is worth adding a discussion with relevant references.
Reply: Thank you! The p62/SQSTM1 (p62) is a selective cargo receptor for misfolded proteins in autophagy (PMID: 31818366). It is reported that p62 binds with Parkin and assist PINK1/Parkin mitophagy pathway (PMID: 28975445). During proteolysis stage of autophagy or mitophagy, the p62 protein itself will also be degraded. Therefore, decreased levels of p62 is a commonly used maker for activated proteolysis (PMID: 19200883). As suggested, we have added these descriptions in Discussion section, please refer to lines 219-226.
- Please add the full name to the some abbreviated terms at the first mention.
Reply: Thank you! The full names for abbreviations were added. Please refer to lines 12-17, 46, 51-53, 59, 100 and 182.

This manuscript is a resubmission of an earlier submission. The following is a list of the peer review reports and author responses from that submission.
Round 1
Reviewer 1 Report
In this report, Hsu et al. aimed to show that arsenic may activate mitophagy to promote the polarization of M1 to M2 cells and that manipulate cytokine production. Firstly, the results shown in this study did not support the main conclusions claimed by the authors. Second, several experiments for autophagy and mitophagy were not appropriately designed. For example, in addition to a little bit increased level of LC3-II in arsenic-treated M1 cells, there was no significant activation of autophagy markers, such as autophagic vacuole formation and/or p62/SQSTM1 degradation shown in the arsenic treatment experiment. For mitophagy induction by arsenic in M1 cells, the authors only provided an upregulated phospho-parkin in arsenic-treated cells and the unclear colocalization between LC3 and mitotracker. In spite of these data, no conventional hallmark of mitophagy was used in this study, such as PINK1 stabilization, ubiquitin phosphorylation, and mitofusin protein degradation. Also, no appropriate control was designed for the assays on phospho-parkin, such as no total expression of parkin shown. In addition, the faint immunofluorescence intensity and resolution did not support the colocalization of LC3 with mitochondria. Third, no gene silencing and/or ectopic expression of autophagy and mitophagy genes were used in this study to demonstrate that autophagy and mitophagy are both required for M1 to M2 polarization and cytokine production. Forth, no compelling information for the materials and methods was provided, such as antibodies used. Fifth, the authors have to critically prepare the content of the submitted manuscript. In general, several conclusions were not supported by the data shown in this study. The authors should be aware of the over-interpretation of results, particularly for autophagy and mitophagy. There were also several typos and grammatic errors in the submitted manuscript.
Author Response
To the Reviewer
We would like to thank you for your comments on our manuscript. We are also grateful that your valuable suggestions will largely improve the quality of our manuscript. However, because we need time to obtain RNAi constructs and antibodies, we could not accomplish all the suggested experiments within 10 days. We hope you would kindly agree us to extend our revision period to 3 weeks (before 9/10).
Best Regards,
Wei-Ting Liao Ph.D.
Reviewer 2 Report
The manuscript is well written, devoted to an interesting subject. The authors used a wide range of methods to obtain results that look representative. However, there are several methodological questions for the authors that will need to be answered in order for the manuscript to be accepted for publication.
11There is no description of PCR protocol in the manuscript. Primer annealing temperatures are not indicated, which should probably vary due to different primer lengths.
22The sequence of some primers for measurement of expression is 32 nucleotides. Can author explain why such long primers were used? Since the average primer length for measurement of expression is from 18 to 25 nucleotides.
33 Author should use standard nomenclature for formatting of gene names (https://www.biosciencewriters.com/Guidelines-for-Formatting-Gene-and-Protein-Names.aspx). In general, for human symbols for genes are italicized (e.g., ARG-1), whereas symbols for proteins are not italicized (e.g., ARG-1). For mice or rats, gene symbols are italicized, with only the first letter in upper-case (e.g., Arg-1). Protein symbols are not italicized, and all letters are in upper-case (e.g., ARG-1). Authors should take this into account in the text of the manuscript, tables and figures.
44Names of antibodies for western blotting and their manufacturers are not indicated in the material and methods.
55The description of statistical analysis is lack in the material and methods
Author Response
To the Reviewers
We would like to thank the reviewers for their comments on our manuscript. We are also grateful to the reviewers for their efforts and time in reviewing the manuscript. These valuable suggestions indeed improved the quality of the manuscript. We have point-to-point revised the manuscript according to the reviewers’ comments. In addition, we have flagged the changes made in the revised manuscript.
Reviewer 2
Comments and Suggestions for Authors
The manuscript is well written, devoted to an interesting subject. The authors used a wide range of methods to obtain results that look representative. However, there are several methodological questions for the authors that will need to be answered in order for the manuscript to be accepted for publication.
- There is no description of PCR protocol in the manuscript. Primer annealing temperatures are not indicated, which should probably vary due to different primer lengths.
Reply: Thank you! The PCR protocol and annealing temperature for each primer are now added in the Material and Methods section. Please refer to page 9, lines 250-253 and the Table for primer sets below.
- The sequence of some primers for measurement of expression is 32 nucleotides. Can author explain why such long primers were used? Since the average primer length for measurement of expression is from 18 to 25 nucleotides.
Reply: We tested several primer sets for testing iNOS mRNA expression, among them, we selected the 32 bp iNOS primers (Jiménez et al., J Virol. 2001; 75: 4655–4663. PMID: 11312336) that generating stable and sensitive experimental results.
- Author should use standard nomenclature for formatting of gene names (https://www.biosciencewriters.com/Guidelines-for-Formatting-Gene-and-Protein-Names.aspx). In general, for human symbols for genes are italicized (e.g., ARG-1), whereas symbols for proteins are not italicized (e.g., ARG-1). For mice or rats, gene symbols are italicized, with only the first letter in upper-case (e.g., Arg-1). Protein symbols are not italicized, and all letters are in upper-case (e.g., ARG-1). Authors should take this into account in the text of the manuscript, tables and figures.
Reply: Thank you very much for this advice! We have revised all the symbols using standard nomenclature in whole manuscript (yellow-labeled in the revised manuscript).
- Names of antibodies for western blotting and their manufacturers are not indicated in the material and methods.
Reply: Thank you! We have listed all antibody details in the Material and Methods section. Please refer to page 10, lines 285-289 and 299-300; page 11, lines 305-308.
- The description of statistical analysis is lack in the material and methods
Reply: The description of statistical analysis has been added in the Material and Methods section. Please refer to page 11, lines 317-320.

Reviewer 3 Report
1Major concerns
figures are not clear. Increase the image qualities
2. Why the M0 has been increased at 5 µM Arsenic
3. Why didn’t use up to 10 µM since 10 µM is not toxic?
4. Is the differentiation protocol obtained from the previous studies? If so, provide the evidence.
5. Italicize the gene names (including in the figures) in 4.3 primers and indicate the amplicon size for future perspectives.
6. If the lysis buffer used for western blotting is a commercial kit, indicate the name. If not indicate the composition.
7. In Fig.1B, M2 polarized macrophages population has been decreased by A5. Justify the results.
8. Why the Arg-1 expression has been decreased in M2 macrophages? Justify the results in the relevant sections.
9. P-Parking band is overexposed. So, the fold changes are not clearly visible. Replace it with a better and clearer one.
10. Fig.3. statistical markings are not visible. Enlarge them.
11. If mitophagy is induced by arsenic in M1 macrophages, that change should be visible in AS1 and AS5 in Fig.4C.
Minor concerns
Line 94, there is a space between µ and M
In Line 312, µm has been written as um
Author Response
To the Reviewers
We would like to thank the reviewers for their comments on our manuscript. We are also grateful to the reviewers for their efforts and time in reviewing the manuscript. These valuable suggestions indeed improved the quality of the manuscript. We have point-to-point revised the manuscript according to the reviewers’ comments. In addition, we have flagged the changes made in the revised manuscript.
Reviewer 3
Comments and Suggestions for Authors
Major concerns
- Figures are not clear. Increase the image qualities.
Reply: The images have been improved. Please refer to revised Figure 1B-D and Figure 4C.
- Why the M0 has been increased at 5 µM Arsenic
Reply: It is known that arsenic has a bi-phase effect on cell viability. In general, lower than 10 µM of arsenic showed growth-promoting effects and higher than 10 µM of arsenic showed cytotoxic effects (Liao et al., Concentration-dependent cellular responses of arsenic in keratinocytes. Kaohsiung J Med Sci. 2011; 27: 390-5. PMID: 21914526; Lee et al., Aberrant cell proliferation by enhanced mitochondrial biogenesis via mtTFA in arsenical skin cancers. Am J Pathol. 2011; 178: 2066-76. PMID: 21514422). In the current study, we observed that 5 µM of arsenic increased cell viabilities in M0, M1 and M2 cells, suggesting non-toxic concentrations of arsenic activated macrophages. However, this increased cell viability were decreased under 10 µM arsenic treatment in M0 cells, but not in M1 and M2 cells. The M0 cells are PMA-treated THP-1 monocytes, which are pre-mature macrophages. It is possible that pre-mature M0 macrophages have lower responses to arsenic, comparing to M1 and M2 macrophages. The above description was added in Discussion section, please refer to page 7, lines 153-160.
- Why didn’t use up to 10 µM since 10 µM is not toxic?
Reply: Chronic arsenic exposure range is between 0.1 to 0.5 mg/kg, approximately equal to equal to 1.4 to 6.9 μM (Kuivenhoven, M.; Mason, K., Arsenic Toxicity. In StatPearls, StatPearls Publishing Copyright © 2022, StatPearls Publishing LLC.: Treasure Island (FL), 2022.). Therefore, we selected 5 µM as our arsenic treatment concentration. Please refer to page 2, lines 44-46.
- Is the differentiation protocol obtained from the previous studies?
If so, provide the evidence.
Reply: Yes. Our differentiation protocol is refer to the study from published study by “Chanput et al., Characterization of polarized THP-1 macrophages and polarizing ability of LPS and food compounds. Food Funct. 2013; 4: 266-76. PMID: 23135314”. We have added this citation in the revised manuscript. Please refer to page 9, lines 233-235.
- Italicize the gene names (including in the figures) in 4.3 primers and indicate the amplicon size for future perspectives.
Reply: Thank you! We have revised the gene names and added the amplicon size for each primer. Please refer to the Table for primer sets in page 9.
- If the lysis buffer used for western blotting is a commercial kit, indicate the name. If not indicate the composition.
Reply: Thanks for your advice. Our lysis buffer is commercial RIPA buffer from Merck. We have added this information in Materials and Methods section. Please refer to page 10, line 279.
- In Fig.1B, M2 polarized macrophages population has been decreased by A5. Justify the results.
Reply: Thank you! Figure 1B showed that arsenic increased CD206 expression in M0 and M1 macrophages, but decreased CD206 expression in M2 macrophages. These description were added in Results section, please refer to page 2, lines 48-52.
- Why the Arg-1 expression has been decreased in M2 macrophages? Justify the results in the relevant sections.
Reply: Arg-1 is a marker gene expressed in M2 macrophages, however, different M2 subtype has different Arg-1 expression levels. It has been reported that Arg-1 is highly expressed in M2a macrophages, and TGF-β1 is highly expressed in both M2a and M2c macrophages (Abdelaziz et al., Alternatively activated macrophages; a double-edged sword in allergic asthma. Journal of translational medicine 2020, 18, 58.). Our results showed that arsenic decreased Arg-1 gene expression but increased TGF-β1 gene expression in M2 macrophages. These results suggested that arsenic promoted M2c polarization in M2 macrophages. The above descriptions were added in Discussion section, please refer to page8, lines 187-191.
The description of arsenic decreased Arg-1 expression in M2 macrophages was added in Results section, please refer to page 4, lines 83-84.
- P-Parking band is overexposed. So, the fold changes are not clearly visible. Replace it with a better and clearer one.
Reply: Thank you! We have improved the Western blotting data. Please refer to the revised figure 4A.
- 3. statistical markings are not visible. Enlarge them.
Reply: Thank you! The statistical markings in Fig. 3 has been enlarged. Please refer to the revised Figure 3.
- If mitophagy is induced by arsenic in M1 macrophages, that change should be visible in AS1 and AS5 in Fig.4C.
Reply: Thank you! As we improved the image resolution, arsenic-induced LC3 accumulation in mitochondria (LC3 and Mitotracker co-localization) is now visible. Please refer to the revised Figure 4C.
Minor concerns
- Line 94, there is a space between µ and M
Reply: Thank you very much! The typo has been corrected. Please refer to line 42.
- In Line 312, µm has been written as um
Reply: The typo has been corrected. Please refer to line 274.

Round 2
Reviewer 3 Report
The authors have dully answered all the concerns raised. I can recommend this for publication. Cheers!